# *Klebsiella pneumoniae* peptide hijacks a *Streptococcus pneumoniae* permease to subvert pneumococcal growth and colonization
Janine Lux[1,2], Hannah Portmann[1], Lucía Sánchez García[1], Maria Erhardt[1], Lalaina Holivololona[1], Laura Laloli[1], Manon F. Licheri[1], Clement Gallay[3], Robert Hoepner[4], Nicholas J. Croucher [5], Daniel Straume [6], Jan-Willem Veening [3], Ronald Dijkman [1], Manfred Heller[7], Denis Grandgirard [1], Stephen L. Leib[1] & Lucy J. Hathaway [1] ✉

Treatment of pneumococcal infections is limited by antibiotic resistance and exacerbation of disease by bacterial lysis releasing pneumolysin toxin and other inflammatory factors. We identified a previously uncharacterized peptide in the *Klebsiella pneumoniae* secretome, which enters *Streptococcus pneumoniae* via its AmiA-AliA/AliB permease. Subsequent downregulation of genes for amino acid biosynthesis and peptide uptake was associated with reduction of pneumococcal growth in defined medium and human cerebrospinal fluid, irregular cell shape, decreased chain length and decreased genetic transformation. The bacteriostatic effect was specific to *S. pneumoniae* and *Streptococcus pseudopneumoniae* with no effect on *Streptococcus mitis, Haemophilus influenzae, Staphylococcus aureus* or *K. pneumoniae*. Peptide sequence and length were crucial to growth suppression. The peptide reduced pneumococcal adherence to primary human airway epithelial cell cultures and colonization of rat nasopharynx, without toxicity. We identified a peptide with potential as a therapeutic for pneumococcal diseases suppressing growth of multiple clinical isolates, including antibiotic resistant strains, while avoiding bacterial lysis and dysbiosis.

Antibiotic resistance is one of the biggest threats to society, with the current trajectory predicted to lead to 10 million deaths due to antibiotic resistance per year by 2050[1]. The economic impact is also significant, with the cost of antibiotic failure estimated to be €1.5 billion in Europe annually[2]. Amongst the priority pathogens listed by the World Health Organization is penicillin-non-susceptible *Streptococcus pneumoniae* (pneumococcus)[3,4]. These Gram-positive bacteria are responsible for diseases including meningitis, pneumonia, and septicaemia with significant morbidity and mortality, particularly in young children, the elderly, and the immunocompromised[5]. The pneumococcus is the most common cause of community-acquired pneumonia globally[5], with

antibiotic resistance a growing concern[6] and so new treatment strategies are needed.

Prior to the potential invasion, *S. pneumoniae* must colonize the human nasopharyngeal niche alongside other members of the microbiota. Some commensal microbiota species such as *Dolosigranulum pigrum, Corynebacterium*[7–9], *Streptococcus salivarius*[10], and *Prevotella melaninogenica*[11] protect from colonization by *S. pneumoniae*, for example, by modulating the innate immune response or by L-lactic acid production. *S. pneumoniae* is also proposed to sense the environment via its transporter Ami-AliA/AliB permease, which takes up oligopeptides, resulting in gene expression modulation[12] and affecting nasopharyngeal colonization[13].

[1]Faculty of Medicine, Institute for Infectious Diseases, University of Bern, Bern, Switzerland. [2]Graduate School for Cellular and Biomedical Sciences, University of Bern, Bern, Switzerland. [3]Department of Fundamental Microbiology, University of Lausanne, Lausanne, Switzerland. [4]Department of Neurology, Bern University Hospital and University of Bern, Bern, Switzerland. [5]MRC Centre for Global Infectious Disease Analysis, Sir Michael Uren Hub, White City Campus, Imperial College London, London, UK. [6]Faculty of Chemistry, Biotechnology and Food Science, Norwegian University of Life Sciences, 1430 Ås, Norway. [7]Proteomics and Mass Spectrometry Core Facility, Department for BioMedical Research (DBMR), University of Bern, Bern, Switzerland. ✉e-mail: lucy.hathaway@unibe.ch

We previously expressed this transporter's oligopeptide binding proteins AmiA, AliA, and AliB as recombinant proteins and identified their peptide ligands from the nasal wash of children. Depending on the peptide, they affected phenotypes such as growth[14,15]. Some of the peptides had sequences that matched ribosomal proteins of *Klebsiella pneumoniae*, another resident of the respiratory tract with the potential to cause pneumonia, and were found in the secretome of *K. pneumoniae*[16]. It is unknown whether a functional Ami-AliA/AliB permease is required for the effect of such *K. pneumoniae* peptides, whether peptide uptake is necessary, and whether other *K. pneumoniae* secretome peptides that hijack the permease remain to be discovered.

To address this, we harvested the extracellular peptidome of actively growing *K. pneumoniae* in a peptide-free medium and analysed it by mass spectrometry. We identified a previously uncharacterized peptide of 11 amino acid length, which inhibited the growth of a wide range of clinical pneumococcal isolates, including those resistant to antibiotics, affected cell shape, decreased chain length, and transformation efficiency. We tested its effects not only in a culture medium but also in human cerebrospinal fluid (hCSF), at the air interface of primary human airway epithelial cell cultures and in vivo in the rat nasopharynx in the presence of a microbiota. Following uptake via the Ami permease, the pneumococcal transcriptome and proteome were altered, and pneumococcal colonization decreased. We propose that this interspecies peptide has therapeutic potential to treat pneumococcal diseases.

## Results

### *Klebsiella pneumoniae* releases a previously uncharacterized ribosomal peptide

We extracted peptides secreted by live *K. pneumoniae* into a chemically defined medium (CDM) by performing solid-phase extraction and identified the peptides by LC–MS/MS. Along with previously identified AmiA and AliA peptide ligands[16], we found another ribosomal peptide of similar amino acid length. Three peptides were present, which included the sequence VNATDEDRWNA, found in ribosomal protein S14: 11-amino acid VNATDEDRWNA, 13-amino acid SDVNATDEDRWNA, and 15-amino acid IISDVNATDEDRWNA. The 11-amino acid peptide is abbreviated here as V11A.

### Peptide V11A suppresses pneumococcal growth

We tested the effect of peptide V11A and the 13- and 15-amino acid versions on the growth of *S. pneumoniae* laboratory strain D39 in CDM. V11A, but not the longer peptides, suppressed growth in a dose-dependent manner (Fig. 1a). By performing BLAST with the sequence of peptide V11A, we found amino acid differences at positions 3, 4, 7, and/or 10 in various species. For example, serine, instead of threonine, at position 4 was found in *Escherichia coli*, *Enterobacter hormaechei*, and *Haemophilus parainfluenzae* strains. We therefore tested whether these differences influence the growth-inhibiting effect of peptide V11A on *S. pneumoniae*. We found that peptides with substitutions at positions 4 or 10 did not suppress pneumococcal growth, whereas substitutions at positions 3 or 7 had no effect (Fig. 1b). Since peptide V11A inhibited the growth of *S. pneumoniae* laboratory strain D39 (Fig. 1a) we tested its effect on diverse clinical isolates. We found that peptide V11A inhibited the growth of a wide range of isolates, regardless of serotype or antibiotic resistance (Table 1 and Supplementary Fig. 1). In contrast to previously identified *K. pneumoniae* ribosomal peptides, AmiA ligand (AKTIKITQTR) and AliA ligand (FNEMQPIVDRQ), V11A suppressed the growth of a trimethoprim-sulfamethoxazole resistant isolate 1154.75 (serotype 23F) (Fig. 1c). Growth of this isolate (1154.75) was also suppressed by V11A in human cerebrospinal fluid (hCSF) from five different donors, mimicking the environment of *S. pneumoniae* during pneumococcal meningitis (Fig. 1d, Supplementary Fig. 2). To test the specificity of growth inhibition by peptide V11A, we tested its effect on growth of other common bacterial colonizers of the nasopharynx. None were inhibited by V11A, except *Streptococcus pseudopneumoniae*, a very close relative of *S. pneumoniae* (Table 1 and Supplementary Fig. 1). There were two pneumococcal strains that were not inhibited by V11A: South African

strain 17619 and laboratory strain R6, which we have shown previously both have mutated AmiA and whose growth is also not suppressed by the AmiA and AliA peptides[16]. A BLAST search of the AmiA sequence covering the relevant mutation was performed to determine its frequency: YGYVY-TADPETLDYLIS**R**K (mutation in bold and underlined). Of the 5000 sequences examined, only 4 (0.08%) had this mutation. A time-kill assay showed that V11A had a bacteriostatic effect on pneumococcus in contrast to the bactericidal antibiotic penicillin G, used as a control (Fig. 1e). In summary, peptide V11A suppressed growth of a range of pneumococcal isolates in CDM and suppressed growth of a clinical pneumococcal isolate in hCSF, the effect was bacteriostatic, species-specific and the peptide length and sequence were critical.

### Pneumococcal growth inhibition by V11A requires uptake via Ami-AliA/AliB permease

As peptide V11A did not inhibit two pneumococcal strains with mutated AmiA we hypothesized that the Ami-AliA/AliB oligopeptide permease is involved in the response to V11A. We confirmed this with a mutant in which AmiC, one of the two permease subunits, was disrupted. In the Δ*amiC* mutant the growth suppression effect of V11A was lost (Fig. 2a). To determine whether peptide V11A is taken up into *S. pneumoniae* D39 and whether a functional Ami-AliA/AliB oligopeptide permease is necessary for this, we incubated D39 and the Δ*amiC* mutant for 5 min with FITC-labelled peptide V11A and performed epifluorescence microscopy. We found clear homogeneous FITC staining intracellularly only in the D39 strain with a functional permease and greatly reduced uptake in the Δ*amiC* mutant (Fig. 2b). In the orthogonal view of a z-stack, we see the FITC staining inside the bacterial cell and not on the membrane (Supplementary Fig. 3). For versions of peptide V11A with amino acid differences that did not suppress pneumococcal growth (positions 4 or 10, Fig. 1b), we found greatly reduced uptake of FITC-labelled peptide in the cytoplasm of either D39 or its Δ*amiC* mutant (Supplementary Fig. 4). In conclusion, we found that pneumococcal growth suppression by peptide V11A requires uptake of the peptide via a functional Ami-AliA/AliB oligopeptide permease.

### Peptide V11A affects pneumococcal morphology, chain length and transformation

The antibiotic aztreonam and β-lactamase inhibitor clavulanic acid have been shown to increase pneumococcal chain length and also induce competence[17], which can allow the pneumococcus to take up and incorporate exogenous DNA, supporting the rapid spread of antimicrobial resistance. We, therefore, analysed the effect of peptide V11A on pneumococcal cell morphology, chain length and transformation rate. We grew three different pneumococcal strains with or without peptide V11A until the mid-log phase and performed microscopy at 40x and 100x magnification. Untreated bacteria within the same chain had homogenous size and shape. In the V11A peptide-treated samples, bacteria within the same chain had heterogenous shapes and sizes, with some cells enlarged and rounder than in the untreated sample (Fig. 3a). For all three pneumococcal strains, peptide V11A also significantly decreased chain length (Fig. 3b, Supplementary Fig. 5). We tested the effect of peptide V11A on competence-stimulating peptide (CSP-1)-mediated transformation rate in strain D39. V11A, but not the control peptide (V11A with an amino acid difference at position 4), reduced the mean transformation rate from 1.44% to 0.169%, which equals an 8.5-fold reduction (Fig. 3c). In summary, we found that V11A changes pneumococcal cell morphology, decreases the chain length and transformation rate.

### Peptide V11A downregulates genes involved in amino acid and protein metabolism

To decipher the mechanism by which V11A caused the phenotypic changes described above, we identified differentially expressed genes in pneumococcal strain D39 treated with peptide V11A compared to the untreated control. Upon V11A treatment, more genes were downregulated than upregulated, as seen in the volcano plot in Supplementary Fig. 6a. Eighty-

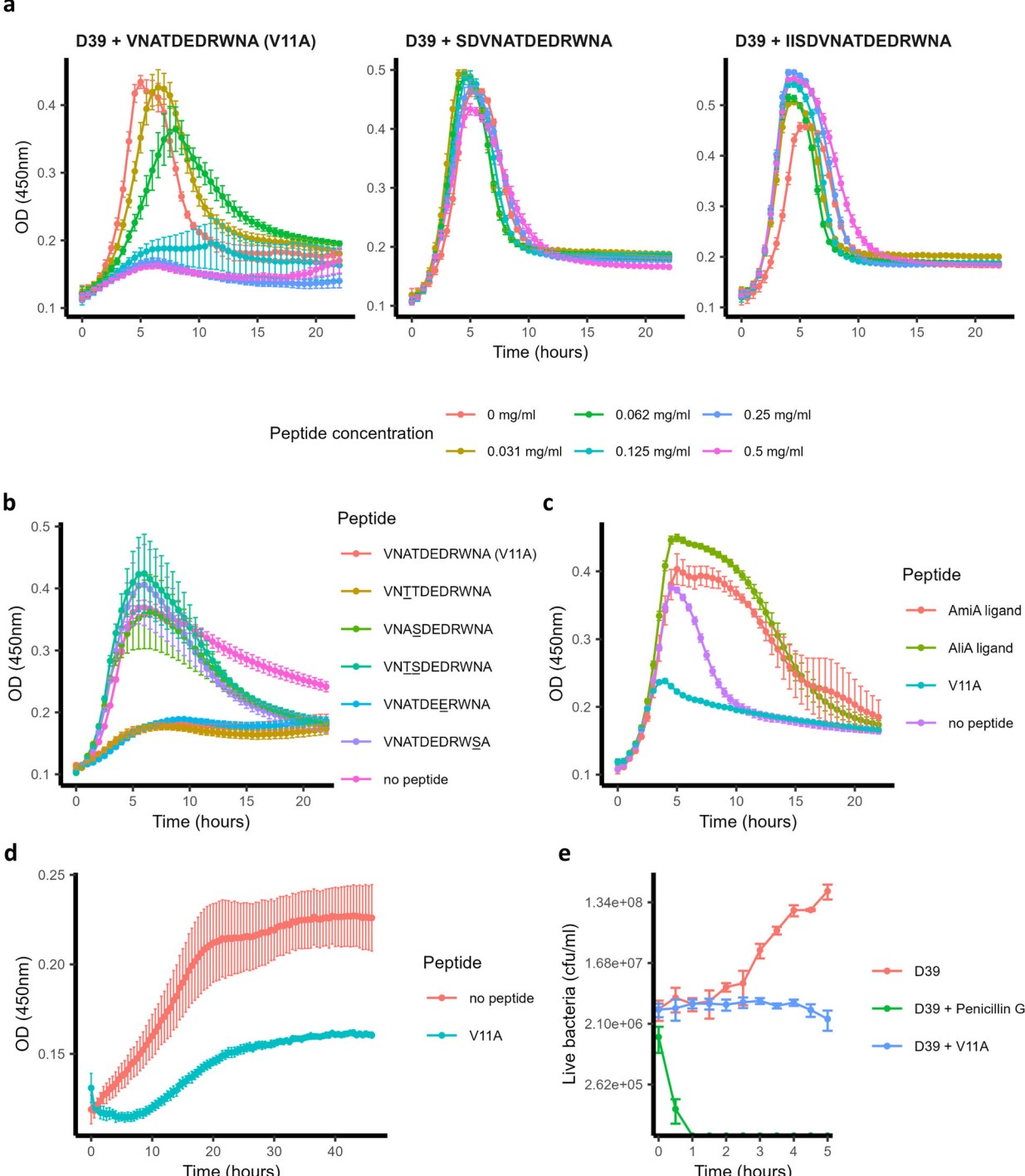

**Fig. 1 | Pneumococcal growth in the presence and absence of peptides. a** Growth curves of pneumococcal strain D39 in the presence and absence of 11 amino acid peptide V11A (VNATDEDRWNA) and its longer versions (13- and 15-amino acid length) in chemically defined medium (CDM). **b** Growth curves of pneumococcal strain D39 in the presence and absence of peptide V11A and peptides with naturally occurring amino acid differences compared to V11A in CDM. Difference at position 3 occurs in strains of *K. pneumoniae* and *Salmonella enterica*, at position 4 in strains of *E. coli*, at positions 3 and 4 in strains of *Arsenophonus*, *E. coli*, *Moritella*, *Pasteurellales*, and *Psychromonas*, at position 7 in strains of *K. pneumoniae* and *S. enterica*, at position 10 in some strains of *Cedecea*, *Psychromonas* and *S. enterica*. Amino acid differences from V11A are underlined. All peptides at a concentration of 0.5 mg/ml. **c** Growth curves of trimethoprim-sulfamethoxazole resistant clinical pneumococal isolate 1154.75 (serotype 23F) in the presence and absence of peptide V11A and *K. pneumoniae* ribosomal peptides AmiA (AKTIKITQTR) or AliA

(FNEMQPIVDRQ) ligand in CDM. **d** Growth curves of pneumococcal strain 1154.75 in the presence and absence of peptide V11A in human cerebrospinal fluid (hCSF) of donor 1 (see Supplementary Fig. 2 for equivalent results in hCSF from four other donors). Bacterial inoculum was twice that used in CDM due to anticipated slower growth in hCSF. **e** Time-kill assay of *S. pneumoniae* strain D39. Bacteria were untreated (negative control) or exposed to 0.5 mg/ml peptide V11A or 0.5 mg/ml penicillin G (positive control for bacteriolytic effect). Increasing cfu/ml represents bacterial growth over time, steady cfu/ml represents bacteriostatic effect, and decreasing cfu/ml represents bacteriolytic effect. Growth was measured by optical density OD (450 nm) measurement in peptide-free CDM for (**a–c**), undiluted human cerebrospinal fluid for (**d**) or by plating dilutions of bacterial sample onto agar plates at the timepoints indicated and counting colony forming units (cfu) for (**e**). Results represent three independent experiments, error bars indicate SEM. Peptide concentration = 0.5 mg/ml unless otherwise indicated.

nine genes underwent a significant change in expression equal or greater than 1.25 log2 fold change (log2FC), of which 64 were downregulated and 25 were upregulated. With these down- and upregulated genes, we performed Gene Ontology (GO) enrichment analysis. For upregulated genes, we did not find biological processes with statistically significant enrichment

**Table 1 | Effect of peptide V11A on the growth of diverse clinical *S. pneumoniae* strains and other bacterial species**

| Species | Strain | Serotype | MIC$_{50}$ V11A (mg/ml) |
|---|---|---|---|
| *S. pneumoniae* | 106.66 (Swiss) | 6B | 0.125 |
| *S. pneumoniae* | 110.58 (Swiss) | Non-typeable | 0.5 |
| *S. pneumoniae* | 1154.75 (Swiss) | 23F | 0.5 |
| *S. pneumoniae* | 1186.70 (Swiss) | 9N | 0.062 |
| *S. pneumoniae* | 1188.31 (Swiss) | 8 | 0.125 |
| *S. pneumoniae* | 1189.53 (Swiss) | 19A | 0.031 |
| *S. pneumoniae* | 1189.65 (Swiss) | 3 | 0.125 |
| *S. pneumoniae* | 1193.68 (Swiss) | 12F | 0.25 |
| *S. pneumoniae* | 208.41 (Swiss) | 7F | 0.25 |
| *S. pneumoniae* | ATCC 17619 (South Africa 19A-7) | 19A | X |
| *S. pneumoniae* | ATCC 700903 (43362 Finland) | 6B | 0.125 |
| *S. pneumoniae* | D39 | 2 | 0.125 |
| *S. pneumoniae* | R6 | Non-typeable | X |
| *S. mitis* | 3308.46 | *n.a.* | X |
| *S. pseudopneumoniae* | 410.05 | *n.a.* | 0.031 |
| *S. aureus* | ATCC 29213 | *n.a.* | X |
| *H. influenzae* | ATCC 49247 | *n.a.* | X |
| *K. pneumoniae* | ATCC BAA-1706 | *n.a.* | X |

MIC$_{50}$ (minimum inhibitory concentration inhibiting at least 50% of bacteria growth) for Swiss clinical *S. pneumoniae* isolates and reference strains, including tetracycline-resistant strain ATCC 700903 with intermediate resistance to penicillin (MIC 1 µg/ml), sulfamethoxazole/trimethoprim-resistant strain 1154.75, erythromycin-resistant strain 106.66 and other bacterial species of the nasopharyngeal microbiota was calculated from data shown in Fig. 1a for D39, Fig. 1c for 1154.75 or Supplementary Fig. 1 for all other strains. "X" indicates a lack of inhibition and "*n.a.*" not applicable.

compared to the genome background. We found several biological processes enriched in the downregulated genes, mainly amino acid biosynthesis, import and transport, but also peptide and protein transport (Supplementary Fig. 6b). With the same cutoffs, we made a gene interaction network (Fig. 4). We see that downregulated genes cluster to the following: AmiA permease (*ami* genes and *aliA*), branched-chain amino acid biosynthesis (*ilv* genes), branched-chain amino acid transporter (*liv* genes) and riboflavin synthesis (*rib* genes). One interesting gene cluster of upregulated genes is fatty acid biosynthesis (*fab* genes). At the level of single genes, the most downregulated gene was SPD_0161, a Mn$^{2+}$/Fe$^{2+}$ transporter and the most upregulated gene was SPD_1524 (transcriptional regulator, *gntR*). We confirmed significant downregulation of *aliA, ilvB, livJ* and significant upregulation of *gntR* by real-time RT-PCR (Supplementary Fig. 6c). We found *codY*, a global regulator of protein metabolism, downregulated −1.2 log2FC. We did not find any *com* genes differentially expressed, however, we found downregulation of other genes potentially involved in competence[18–20]: *ssb* with −0.5 log2FC, *hrcA* with −1.1 log2FC and *grpE* with −1.2 log2FC. We found *dnaJ* and *dnaK* downregulated with −1.6 log2FC and −1.3 log2FC, respectively. *Lic* genes (*licC, licD1-D3, licT*) were downregulated by −0.3 to −0.7 log2FC, *phtD, phtE* downregulated −1.1 and −1.9 log2FC, respectively. *CbpA* was downregulated −0.5 log2FC and *nanA* -1 log2FC. To summarize, we found that V11A altered the pneumococcal transcriptome, mainly downregulating genes involved in amino acid and protein metabolism, but also transcriptional regulators.

**Peptide V11A alters the pneumococcal proteome**

We also analysed the effect of peptide V11A on the proteome of *S. pneumoniae* strain D39. We identified some proteins only detected in the sample without peptide treatment as "off" and proteins only detected in the peptide treatment sample as "on" in Table 2 for *S. pneumoniae* strain D39 and in Supplementary Tables 1a and 1b for strains 106.66 and 208.41. We found choline-binding protein D to be switched off by V11A treatment and dehydratase FabZ to be turned on in strain D39. In strain 106.66, we found the permease protein of the branched-chain amino acid ABC transporter (encoded by the *livH* gene) and riboflavin biosynthesis protein RibBA to be turned off. Thus, we found alteration of the proteome by V11A with some proteins turned on or off in line with the transcriptomic changes.

**No toxic effect was detected for peptide V11A to human airway epithelial cell cultures or zebrafish larvae**

Well-differentiated airway epithelial cell (AEC) cultures are organotypic cell cultures with air–liquid interfaces and recapitulate many aspects of the

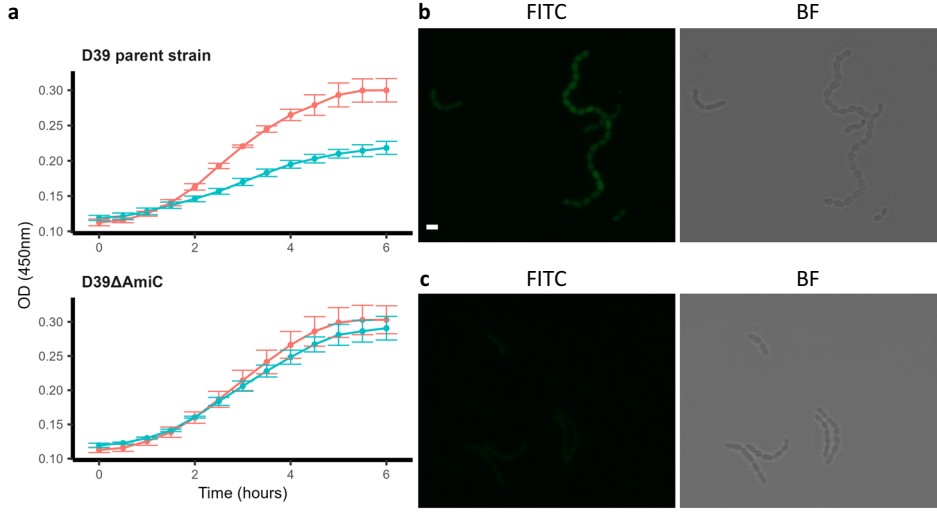

**Fig. 2 | Peptide V11A uptake via Ami permease.**
**a** Growth curve of *S. pneumoniae* D39 parent strain and Δ*amiC* mutant. Growth was measured in peptide-free chemically defined medium (CDM) in the presence and absence of 0.5 mg/ml peptide V11A by measuring optical density OD (450 nm) over time. Results represent three independent experiments, error bars indicate SEM.
**b**, **c** Microscopy images of *S. pneumoniae* after incubation with FITC-labelled peptide V11A for D39 parent strain (**b**) and Δ*amiC* mutant (**c**). Representative images were taken at mid-bacterium localization in the *z* position showing FITC and Brightfield (BF) channels. Scale bar indicates 2 µm for all pictures in (**b**) and (**c**).

**Fig. 3 | Effects of V11A on pneumococcal morphology, chain length and transformation.**
**a** Microscopy images of pneumococci to assess pneumococcal cell morphology. Representative images of *S. pneumoniae* strains D39, 106.66, 208.41 after incubation with or without 0.5 mg/ml peptide V11A for 2.5 h taken at 100x magnification. Scale bar indicates 2 μm for all images. **b** Chain length of *S. pneumoniae* strains D39, 106.66, 208.41. Results represent chain length measurements of three independent experiments after incubation with or without 0.5 mg/ml peptide V11A for 2.5 h, each box represents length measurements of >1200 bacterial chains, *** indicates *p* value < 0.001 by two-sided Wilcoxon rank sum test for all comparisons.
**c** Transformation rate of *S. pneumoniae* strain D39 without or with 0.5 mg/ml peptide V11A or control peptide (V11A with amino acid difference at position 4). Results represent at least five independent experiments for each condition represented by grey dots, **** indicates *p* value ≤ 0.0001 by *t*-test.

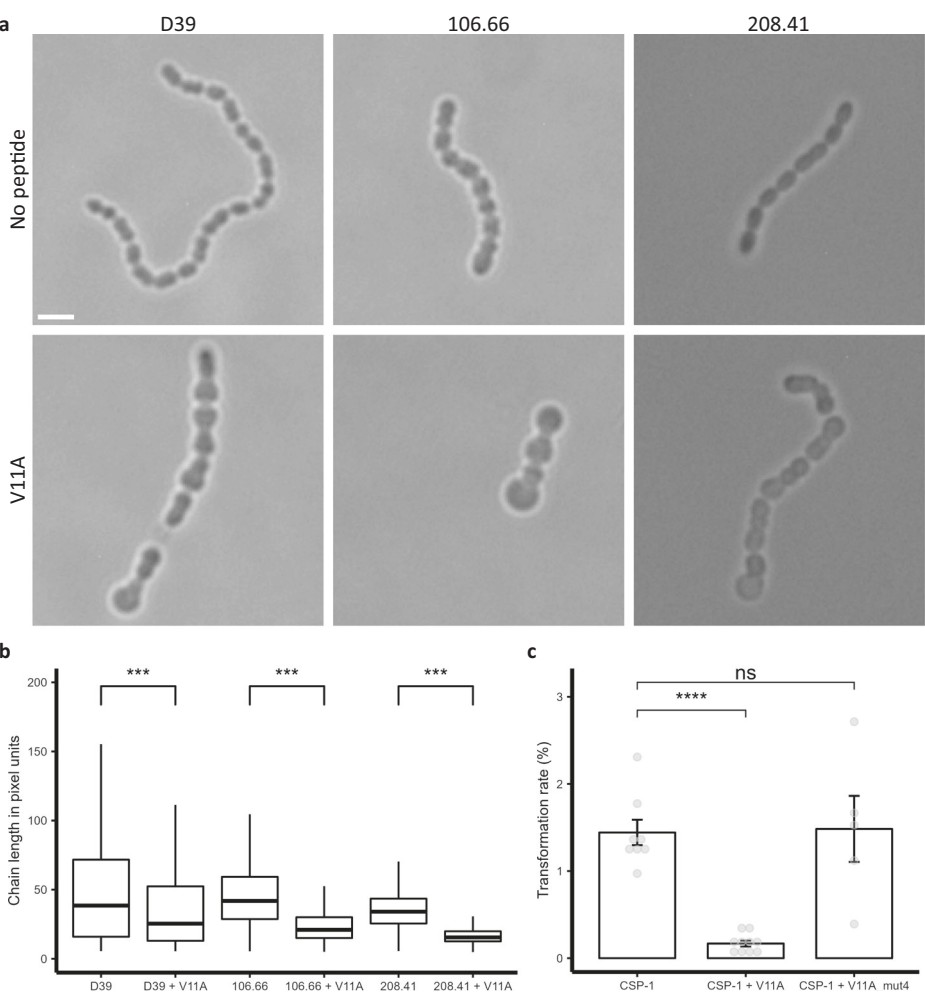

respiratory tract, including airway cell heterogeneity, mucus production and ciliary beating activity[21]. To evaluate whether peptide V11A is cytotoxic to primary human AEC (hAEC) cultures, we incubated the hAEC cultures together with peptide V11A for 30 h. We collected the medium from the apical and basolateral side and determined cytotoxicity calculated as %LDH release using the formula of Rayamajhi et al.[22]. The LDH release was the same and low in media collected from hAEC cultures treated with and without peptide V11A. Therefore no cytotoxicity was detected (Supplementary Fig. 7a). Immunofluorescence staining also shows the same cell morphology with intact tight junctions and ciliated cells present in all samples (Supplementary Fig. 7b). To test whether the peptide is toxic in vivo, zebrafish larvae were exposed to 0.5 mg/ml or 0.25 mg/ml V11A and their swimming behaviour determined in light and dark conditions. This light–dark locomotion test is a well-established metric for toxicity measurements that can reveal, for example, reactivity and muscular weakness[23,24]. The presence of the peptide did not affect the swimming behaviour (Supplementary Fig. 7c). In conclusion, no toxic effect of peptide V11A was detected in vitro to hAEC cultures or in vivo to zebrafish larvae.

## Peptide V11A inhibits pneumococcal adherence to human airway epithelial cell (hAEC) cultures and colonization of the rat nasopharynx

Increased chain length has been associated with increased adherence of pneumococci to epithelial cells[25]. As peptide V11A decreased pneumococcal chain length (Fig. 3b, Supplementary Fig. 5) and was not toxic to hAEC cultures (Supplementary Fig. 7), we tested its effect on pneumococcal adherence in vitro using the same model. We exposed hAEC cultures from

three different donors to *S. pneumoniae* strain D39 and quantified the number of adhered bacteria on the apical surface after removing non-adhered bacteria 24 h post-inoculation. Peptide V11A, but not the control peptide (V11A with an amino acid difference at position 10), decreased bacterial adherence to 22% of that of the control without peptide (Fig. 5a). The control peptide had no effect on adherence. To test whether peptide V11A had an effect in vivo in the presence of the microbiota, we inoculated rat pups intranasally with *S. pneumoniae* strain 1154.75 (serotype 23F) with or without peptide V11A and quantified bacterial load in nasal wash and nasopharynx 24 and 48 h later. At 24 h, pneumococci were detected in the nasopharynx of 9 out of 11 rats (81.81%) in the absence of V11A and in 3 out of 11 (27.27%) in the nasopharynx of rats that had received V11A (Chi-squared test: *p* = 0.03228). Bacterial loads at 24 h are shown in Fig. 5b and were significantly lower in the nasopharynx of rats that had received V11A (*p* = 0.01367). Fewer bacteria in the nasopharynx of the peptide-treated rats than controls were also seen at 48 h (not significant, *p* = 0.30, Supplementary Fig. 8a). Bacterial load in the nasal wash was also reduced significantly in presence of V11A at 24 h (*p* = 0.01989) and 48 h (*p* = 0.01704) (Supplementary Fig. 8b). Therefore, peptide V11A inhibited pneumococcal adherence in vitro to hAEC cultures and colonization in vivo of the rat nasopharynx.

## Discussion

We found a previously uncharacterized peptide, V11A, in the secretome of *K. pneumoniae*, which modulates pneumococcal phenotypes, particularly growth and colonization. For the first time we show that an interspecies peptide suppresses pneumococcal growth not only in a defined medium but in human cerebrospinal fluid, a site of infection in pneumococcal

**Fig. 4 | Effect of peptide V11A on pneumococcal transcriptome.** Genetic interaction network of significantly downregulated (log2FC ≤ −1.25) and upregulated (log2FC ≥ 1.25) genes in *S. pneumoniae* strain D39 after 15 min of incubation with 0.5 mg/ml peptide V11A. Interactions between 2 genes (nodes) are depicted as a line (edge) and were imported from the STRING database. Nodes are colour-coded according to the log2FC, with blue representing negative log2FC (downregulated) and red positive log2FC (upregulated). We highlighted Gene Ontologies or keywords of groups of genes in the background.

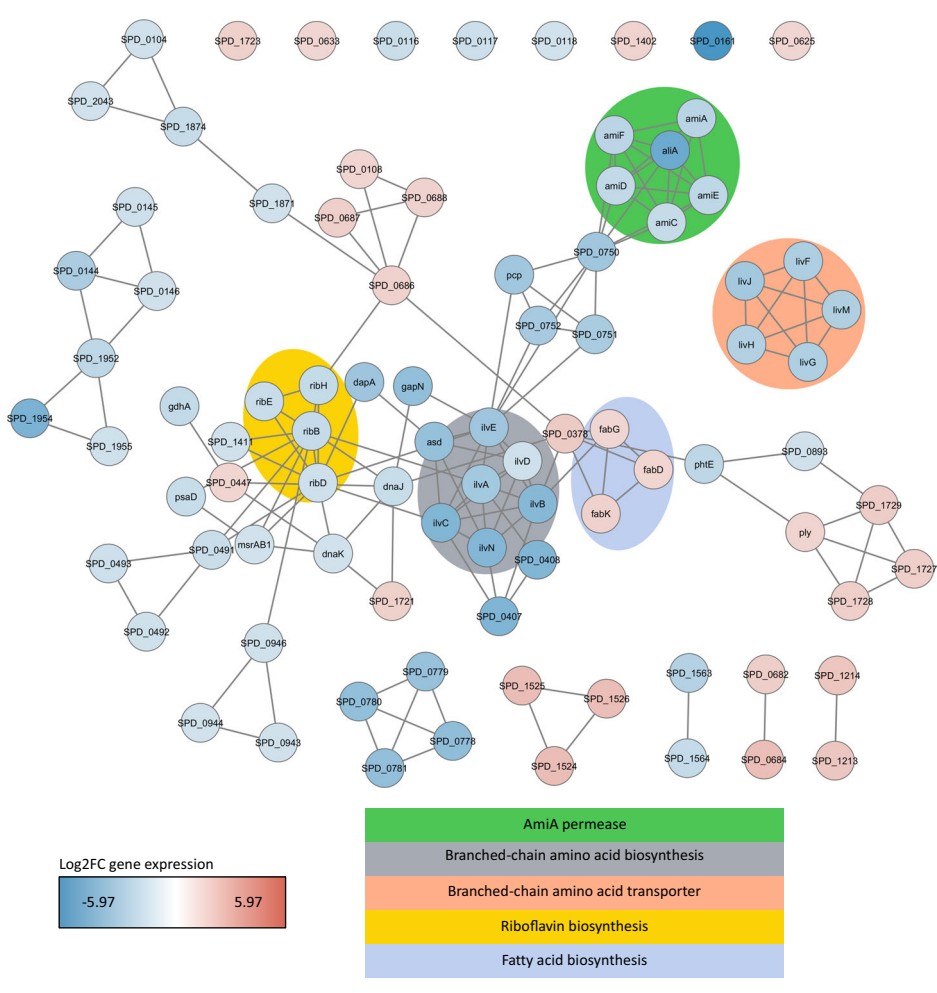

meningitis. Growth suppression is achieved following uptake via the Ami-AliA/AliB permease leading to transcriptional and proteomic changes.

V11A inhibited pneumococcal growth in a dose-dependent manner, with the effect lost for versions of the peptide longer than 11 amino acids or with differences at positions 4 or 10. These changes presumably prevent the peptide from binding to its target on the pneumococcus. V11A suppressed the growth of genetically diverse clinical isolates of pneumococcus, which bodes well for its use as a therapeutic agent. Two strains unaffected by V11A are known to have mutations in the protein AmiA which led us to speculate that AmiA is required for the effect of V11A[16]. The role of the Ami-AliA/AliB oligopeptide permease in the uptake of V11A was confirmed by the observation that deletion of one of the permease subunits, AmiC, abrogated the effect of the peptide on growth and that fluorescently labelled V11A was taken up by the D39 parent strain, but not the Δ*amiC* mutant. V11A did not affect the growth of other bacterial colonizers of the nasopharynx tested: *S. mitis*, *H. influenzae*, *S. aureus* or *K. pneumoniae*. Another proteinaceous interspecies molecule from the genus *Lysinibacillus* was found previously, which depends on the Ami permease for growth inhibition of not only *S. pneumoniae* but also *S. mitis*[26]. V11A therefore appears to have a more species-specific effect. This specificity towards pneumococci is an attractive feature and an advantage over antibiotics which affect multiple species and may lead to dysbiosis of the microbiota. This is also an advantage of V11A over antimicrobial peptides (AMPs) of the host innate immune response, which tend to have a broad effect[27]. In contrast to β-lactam lytic antibiotics, such as penicillin, used to treat pneumococcal infections, V11A had a bacteriostatic effect. By reducing the number of bacteria but not causing their lysis, little pneumolysin should be released compared to treatment with a β-lactam antibiotic. This would be a significant advantage in treatment,

particularly of meningitis where antibiotic-induced bacterial lysis causes the release of pneumolysin, which exacerbates the inflammatory response, neuronal death, and therefore disease[28,29]. The host cytokine response to bacterial lysis can reduce the chance of survival and increase neurological sequelae such as hearing loss and learning difficulties in patients who do survive[30].

We can associate many of the phenotype changes observed with changes in the transcriptome or proteome caused by peptide V11A. The genes most downregulated by peptide V11A were all associated with transport, energy or the synthesis and transport of branched-chain amino acids. Efficient acquisition of branched-chain amino acids is associated with nasopharyngeal colonization[31]. The top-upregulated gene was *gntR*, which has a role in sensing environmental and nutritional cues[32].

The proteins FabK, FabG and FabF, which are encoded in the *fabKDGF-accB-fabZ-accCDA* operon, participate in enzymatic reactions of fatty acid biosynthesis by adding malonyl-coenzyme-A molecules to the growing hydrocarbon chain[33]. We found FabZ protein expression switched on in the presence of peptide V11A along with heterogeneous cell sizes and decreased chain length. *Fab* and *acc* genes were also upregulated in the transcriptome following peptide exposure. Another study found that a competence-associated quorum peptide, BriC, altered fatty acid biosynthesis in *S. pneumoniae* and also promoted nasopharyngeal colonization[34]. We found modulation of fatty acid biosynthesis by interspecies peptide V11A. This, together with observed phenotype changes, is in agreement with a study using a CRISPRi screen where repression of *fabD, fabK, fabZ, accB* and *accD* led to irregular cell shapes and longer chains[35]. Cell chaining is associated with a longer competence and transformation time window as cell chains retain, rather than diffuse the CSP peptide[17]. This is compatible

with our finding that in addition to decreased chain length, we also found that peptide V11A decreased the transformation efficiency of *S. pneumoniae*. Competence contributes to the astonishing genomic plasticity of *S.*

**Table 2 | Proteomic changes detected in *S. pneumoniae* strain D39 after 15 min of exposure to 0.5 mg/ml peptide V11A**

| Protein | Gene | Description |
|---|---|---|
| off | | |
| A0A0H2ZMI0 | SPD_1947 | Transcriptional regulator, putative |
| A0A0H2ZNK9 | SPD_1651 | Iron-compound ABC transporter, ATP-binding protein |
| A0A0H2ZNQ1 | *riml* | [Ribosomal protein S18]-alanine N-acetyltransferase |
| A0A0H2ZP12 | *cbpD* | Choline binding protein D |
| A0A0H2ZPD7 | SPD_0908 | Sua5/YciO/YrdC/YwlC family protein |
| A0A0H2ZQ49 | SPD_0072 | Glyoxalase family protein |
| A0A0H2ZRC4 | SPD_0846 | Membrane protein, putative |
| on | | |
| A0A0H2ZM42 | SPD_0923 | Uncharacterized protein |
| A0A0H2ZN26 | SPD_1594 | Transcriptional regulator |
| A0A0H2ZPR6 | SPD_1517 | Uncharacterized protein |
| A0A0H2ZPT8 | *rafE* | Sugar ABC transporter, sugar-binding protein |
| A0A0H2ZR72 | SPD_1833 | PTS system, IIA component |
| Q04M51 | *fabZ* | 3-hydroxyacyl-[acyl-carrier-protein] dehydratase FabZ |

Proteins that were only detected in the absence of peptide ("off") or only detected in presence of peptide ("on").

*pneumoniae*, which can facilitate antigenic variation, vaccine escape and the acquisition of antibiotic resistance[19]. For a potential therapeutic agent, decreasing the ability of the pneumococcus to acquire antibiotic resistance is of advantage. We did not find any *com* genes, which are involved in regulating and developing competence for genetic transformation[36], differentially expressed upon peptide V11A treatment to explain the reduction of transformation efficiency. However, we found downregulation of other genes, namely *ssb*, *hrcA*, *grpE*, *dnaK* and *dnaJ*, which have been associated with regulating competence[18,20]. The *ssbB* gene is regulated by the alternative sigma factor ComX[18] and therefore expressed uniquely during competence for genetic transformation. It encodes an alternative single-stranded DNA-binding protein that might increase the likelihood of multiple transformation events in the same cell[37]. The pneumococcal competence response exhibits a broad phenotype, combining chaperone and protease production with genetic recombination, which includes induction of such stress response proteases and chaperones[38]. Furthermore, in the proteomics data, CbpD was turned off by V11A peptide treatment. CbpD is a competence-specific murein hydrolase that lyses a subfraction of the pneumococcal population or close relatives in order to release DNA for uptake[39,40] and is produced exclusively by competent cells[41]. One study found that deletion of CbpD reduced transformation efficiency in *Streptococcus thermophilus*[42]. The same has been observed in *Streptococcus sanguinis*[43] lacking LytF and *Streptococcus suis*, lacking CrfP[44], which are both functional analogues of CbpD.

The CodY regulon is involved in the regulation of genes involved in amino acid metabolism, carbon metabolism and iron uptake and among genes described as differentially expressed in D39Δ*codY* were also *aliA, ilv* and *liv* genes[45]. In our transcriptomics data *aliA, ilv* and *liv* genes and *codY* are differentially expressed upon peptide V11A treatment.

Colonization, a prerequisite for nasopharyngeal carriage of *S. pneumoniae*, allows pneumococcus to spread between hosts and to cause invasive disease. Adherence to host cells is one of the main features facilitating colonization[46]. The Ami permease has been suggested previously to

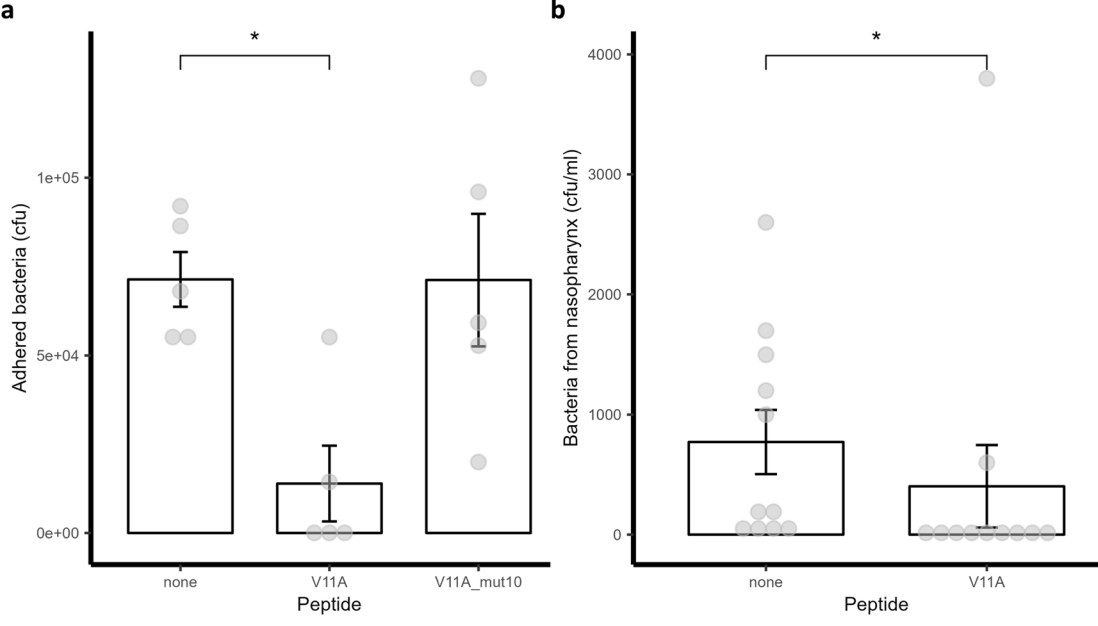

**Fig. 5 | Effect of peptide V11A on adherence of *S. pneumoniae* in vitro and colonization in vivo. a** Pneumococcal adherence to primary human airway epithelial cell (hAEC) cultures. *S. pneumoniae* strain D39 was added to the apical side of differentiated hAEC cultures without or with 0.5 mg/ml peptide V11A or control peptide (V11A with an amino acid difference at position 10). Results represent 5 biological replicates using hAEC cultures of 3 donors on one and 2 donors on another day for each condition, respectively. *indicates *p* value = 0.011 by pairwise one-sided Wilcoxon rank sum test. **b** Pneumococcal adherence to rat nasopharynx.

Trimethoprim-sulfamethoxazole-resistant clinical pneumococcal isolate 1154.75 (serotype 23F) was administered intranasally to rat pups without or with 0.5 mg/ml peptide V11A. Results represent a total of 11 individual pups per condition (from 2 different days with 6 and 5 pups, respectively). * indicates *p* value = 0.01367 by pairwise one-sided Wilcoxon rank sum test. Adhered bacteria to hAEC cultures or rat nasopharynx were quantified 24 h post-inoculation by dilution plating and counting colony-forming units (cfu). Data points are represented by grey dots, error bars indicate SEM.

modulate pneumococcal adherence to epithelial cells by modulating the expression of adhesins on the pneumococcal surface during the first stages of colonization[47]. Increased bacterial chain length has been associated with enhanced adherence of pneumococci to epithelial cells[25]. In accordance with this, peptide V11A, which decreased chain length, inhibited pneumococcal adherence to human airway epithelial cell cultures. V11A downregulated *phtD*, *phtE*, *cbpA* and *nanA*. Pneumococcal surface and histidine triad proteins PhtD and PhtE play a role in adherence to respiratory mucosa and anti-PhtD antibodies can inhibit bacterial attachment[48,49]. CbpA binds the host proteins factor H and vitronectin[46], which might facilitate adherence to host cells and NanA has been suggested to function as an adhesin[50]. Furthermore, V11A downregulated *lic* genes, including *licC*, *licD1-D3*, *licT*. LicD2 mutants showed decreased transformation competence, decreased adherence, reduced nasopharyngeal colonization and reduced virulence in another study[51].

To have potential as a therapeutic, the peptide must have an effect in vivo. We found that V11A significantly reduced the percentage of rats colonized and bacterial load in the nasopharynx and nasal wash of infant rats, indicating its effectiveness in a nutritionally complex environment in the presence of a microbiota.

V11A also reduced the growth of isolate 1154.75 (serotype 23F), which the AmiA and AliA ligands did not. Furthermore, AmiA and AliA peptides increased the chain length of bacteria and did not affect adherence to epithelial cells[15], while V11A decreased both chain length and adherence. While the AmiA and AliA peptides reduced the transformation rate by 3-fold[15], V11A caused an 8.5-fold reduction.

The limitations of the study include that we used high doses of peptides, but optimization, for example, of delivery, could reduce effective doses. Future studies would also need to test whether resistance to peptides develops over time, as happens with antibiotics. Alterations to the genes encoding penicillin-binding proteins, involved in cell wall construction, allow bacteria to escape the action of the β-lactam antibiotics[52,53]. However, the peptide acts by specific binding to the substrate-binding protein of an ABC transporter which has a role in sensing peptides in the environment. Mutation would mean the loss of this function. Therefore, we predict this is less likely to occur. This is supported by the fact that a BLAST search of AmiA and AliA proteins indicates that they are highly conserved[16] and the mutation in AmiA of strain R6, associated with lack of response to V11A, was found in fewer than 0.08% of pneumococcal genomes.

A limitation of the colonization study is that peptide was administered simultaneously with the bacteria, in contrast to the situation of treatment when colonization/infection would be established then peptide administered. However, the experiment established proof of concept that peptides can be effective in vivo.

We have shown that interspecies peptide V11A produced by *K. pneumoniae* modulated phenotypes of *S. pneumoniae*, particularly suppressing its growth in the medium and in human cerebrospinal fluid. We speculate that the import of the peptide via the pneumococcal AmiA-AliA/AliB permease may be advantageous to *K. pneumoniae* in the shared niche of the nasopharynx.

Uptake of peptide V11A led to the downregulation of genes for amino acid biosynthesis, caused irregular cell shape, decreased chain length and decreased genetic transformation. The bacteriostatic effect was species-specific and dependet on peptide sequence and length. The peptide reduced pneumococcal adherence to hAEC cultures and colonization of rat nasopharynx, without apparent toxicity. We, therefore, propose that peptide V11A has potential as a therapeutic agent against pneumococcal diseases with the advantages of effectiveness against antibiotic-resistant strains, avoiding bacterial lysis and the subsequent damaging inflammatory response and specificity of action, avoiding dysbiosis of the microbiota.

## Methods
### Bacterial strains and culture conditions
Bacterial strains and culture conditions were used as described previously[16]. Briefly, the bacterial strains were stored at -80 °C in the Protect

Microorganism Preservation System (Technical Service Consultants Ltd.). As previously described, *S. pneumoniae* was streaked out on CSBA plates, *H. influenzae* on CHOC plates and both grown overnight at 37 °C and 5% $CO_2$; *K. pneumoniae* and *S. aureus* were streaked out on CSBA plates and grown overnight at 37 °C and atmospheric $CO_2$ concentration. Swiss clinical *S. pneumoniae* strains originated from a strain collection obtained from nationwide surveillance for nasopharyngeal *S. pneumoniae* isolates from children with respiratory infection[54] or were kindly provided by Carlo Casanova, Institute for Infectious Diseases, Bern. Strains used, including reference strains, are shown in Table 1. The D39 parent strain (ds865, D39 (Δ*comAB::erm*)) and its mutant Δ*amiC* (ds1039, D39, Δ*amiC::Janus*) were used in Fig. 2[26]. Swiss clinical isolates of commensal streptococci *S. pseudopneumoniae* (410.05) and *S. mitis* (3308.46) were also used[16]. As previously, chemically defined media (CDM)[16], BHI medium and PBS were made in-house.

### Identification of peptides in the secretome
Peptides secreted by *K. pneumoniae* were identified as described previously by mass spectrometry[16].

### Peptides
Synthetic peptides (Genscript, Thermo Fisher Scientific) were ordered with ≥95% purity. Peptides for fluorescent microscopy were labelled at the C-terminus with Lys(FITC). For human airway epithelial cell cultures and in vivo experiments (zebrafish larvae, rats), peptides were endotoxin-free and with TFA removal (salt form: acetate).

### Human cerebrospinal fluid
Residual human cerebrospinal fluid (hCSF) from five anonymized patients undergoing lumbar puncture in 2023 due to non-inflammatory conditions was used for growth assay.

### Growth assay
Growth assays were done as described previously[16] and plotted in R with error bars representing SEM of 3 independent biological replicates. For growth in hCSF, bacteria were inoculated as described elsewhere[55] with twice the inoculum in PBS compared to the growth assay in CDM due to anticipated slower growth in hCSF.

### Time-kill assay
Following overnight culture on agar plates, bacteria were sub-cultured in BHI medium overnight for approximately 7 h until the cultures reached $OD_{600nm} = 0.5$ in mid-log phase. 500 μl of the overnight culture were sub-cultured in 5 ml of fresh BHI until $OD_{600nm} = 0.5$, centrifuged at 3000×*g* for 7 min and the pellet resuspended in 5 ml CDM. The absorbance was adjusted in CDM to $OD_{600nm} = 0.05$, then 3 ml of this bacterial suspension was mixed with 7 ml CDM to reach ca. $5 \times 10^6$ cfu (colony forming units)/ml. Penicillin G and peptide were resuspended to 0.5 mg/ml with the bacterial suspension. In Eppendorf tubes, 1 ml cultures without or with Penicillin G or peptide were grown in a water bath at 37 °C and every 30 min for 5 h, 10 μl were removed to plate out dilutions onto CSBA plates. Viable bacteria counts were determined after overnight incubation of the CSBA plates at 37 °C and 5% $CO_2$.

### Epifluorescence microscopy
Following overnight culture on agar plates, *S. pneumoniae* D39 parent strain and D39Δ*amiC* were grown in BHI to $OD_{600nm} = 0.1–0.2$, 1 ml culture centrifuged at 10,000×*g* for 1 min and the pellet resuspended in 100 μl BHI and 50 μl of 1 mg/ml fluorescently labelled peptide. After 5 min incubation time, the bacteria were washed in 1 ml PBS three times. After the last wash, the pellet was resuspended in 40 μl of PBS and 1.5 μl was spotted onto an acrylamide pad in a gene-frame on a microscope slide. We prepared acrylamide slides by sticking a gene-frame (Thermo Scientific, AB0578) onto a glass slide and mixing the following in a tube per acrylamide slide: 370 μl PBS, 125 μl 40% acrylamide/bis-acrylamide 29:1 (Sigma-Aldrich, A7802),

0.5 µl TEMED (Sigma-Aldrich, T9281), 5 µl freshly prepared 10% APS (Sigma-Aldrich, A3678). From this mix, we poured 500 µl into the gene-frame and added another glass slide to close it. After 30 min at room temperature, we gently slid off one glass slide, cut the acrylamide pads into small pieces and incubated them in PBS for at least 30 min before use.

Microscopy was performed using a Zeiss Axio Imager M1 fluorescence microscope with a ×100 oil immersion objective (EC Plan-Neufluoar 100x/1.30 Oil M27), filter with 493 nm excitation and 517 nm emission wavelength for the FITC channel and images photographed by a Zeiss Axiocam 712 mono camera. We edited pictures for Fig. 2b with the same settings for all samples to maintain comparability between samples and intensity of fluorescence using the Software Zeiss ZEN 3.8: denoising the brightfield channel picture with setting real wavelets to 1.0, followed by adjusting the histogram upper white limit in the brightfield channel to 6000 and in the FITC channel to 5000.

### Bacterial shape characterization

Bacterial culturing was done as described for the growth assay for 2.5 h then 800 µl were centrifuged at $3000 \times g$ for 7 min. Most of the supernatant was discarded and the pellet resuspended in the remaining volume of CDM (ca. 30 µl). A bacterial suspension of 10 µl was mixed with 2 µl fluorescein isothiocyanate (FITC)-Dextran solution (Sigma-Aldrich, resuspended to 10 mg/ml in water). Of this mixture, 10 µl were pipetted onto a microscope slide and the coverslip was applied firmly. The slides were viewed using a Zeiss Axio Imager M1 fluorescence microscope with a ×100 oil immersion objective and brightfield images photographed by a Zeiss AxioCam HRc camera.

### Chain formation

Bacteria were cultured as described above and 8 µl were pipetted onto a microscope slide, air-dried, heat-fixed by flame and Gram-stained using the Aerospray TB Slide Stainer (ELITechGroup). Pictures were taken with ×40 objectives in the Cytation 5 multimode reader (Biotek) with the Gen5 Software (Biotek). Microscopy image analysis of chain length was done using MicrobeJ[56]. Results for each sample represent measurements of >1200 bacterial chains from at least 12 pictures of 3 independent biological replicates. Bacteria chain detection settings (=objects) were set to exclude objects on the picture edge, objects with branching and only recognized as objects with a pixel width of 5-15.

### Transformation assay

The protocol was adapted from elsewhere[57] to the following: *S. pneumoniae* strain D39 was grown in BHI + 5% FCS to $OD_{600nm} = 0.5$, then 150 µl sub-cultured in 9.5 ml of fresh BHI + 5% FCS until $OD_{600nm} = 0.1$. After centrifugation at $3000 \times g$ for 7 min the pellet was resuspended in the same volume of CDM. One ml of the bacterial suspension with 100 ng/ml CSP-1 (EMRLSKFFRDFILQRKK) with and without 0.5 mg/ml peptide V11A was incubated for 12 min at 37 °C. After 100 µl were transferred into a new prewarmed tube and 1 µg chromosomal DNA from streptomycin-resistant strain 104.37 was added, and the culture was incubated for 20 min at 30 °C, then 900 µl prewarmed CDM was added, and the culture incubated for 90 min at 37 °C. Serial dilutions in PBS were plated onto CSBA plates with and without 200 µg/ml streptomycin. The number of colonies was counted after overnight incubation and the transformation rate was calculated.

### Gene expression analysis by RNA-Seq and RT-qPCR

Following overnight culture on agar plates, *S. pneumoniae* strain D39 was grown in BHI + 5% FCS to $OD_{600nm} = 0.5$ and 100 µl of the overnight culture was sub-cultured in 10 ml of fresh CDM until $OD_{600nm} = 0.2$. The culture was split into two 5 ml cultures and 0.5 mg/ml peptide was added to one sample. After 15 min incubation at 37 °C, transcription was stopped with RNAprotect bacteria reagent (Qiagen) and total RNA was isolated with the RNeasy kit (Qiagen) according to the manufacturer's manual. RNA concentration and integrity were assessed with Nanodrop (Thermo Scientific) and Agilent Biolanalyzer (Agilent) before submitting 6 biological

replicates of each sample to the Next Generation Sequencing Platform (University of Bern), where quality control assessments, rRNA depletion, generation of libraries and sequencing were carried out. At the Next Generation Sequencing Platform (University of Bern), the quantity and quality of the purified total RNA were assessed using a Qubit 4.0 fluorometer (Thermo Fisher Scientific) with the Qubit RNA BR Assay Kit (Thermo Fisher Scientific, Q10211) and an Advanced Analytical Fragment Analyser System using a Fragment Analyser RNA Kit (Agilent, DNF-471), respectively. Thereafter, 150 ng of input RNA was depleted of ribosomal RNA using a RiboCop rRNA Depletion Kit for Gram-Positive Bacteria (G+) (Lexogen, SKU 127) following the Lexogen User guide 125UG246V0102. Next, cDNA libraries were generated using a CORALL RNA-Seq V2 Library Prep Kit with UDI 12 nt Set A3 (Lexogen, SKU 173) according to the RTL protocol and 15 PCR cycles (Lexogen User Guide 171UG394V0100). The resulting cDNA libraries were evaluated using a Qubit 4.0 fluorometer with the Qubit dsDNA HS Assay Kit (Thermo Fisher Scientific, Q32854) and an Agilent Fragment Analyser with an HS NGS Fragment Kit (Agilent, DNF-474), respectively. Pooled cDNA libraries were sequenced paired-end using a NextSeq 1000/2000 P2 Reagents v3 (200 cycles; Illumina, 20046812) on an Illumina NextSeq 1000 instrument. The run produced, on average, 26 million reads/library. The quality of the sequencing run was assessed using Illumina Sequencing Analysis Viewer (Illumina version 2.4.7) and all base call files were demultiplexed and converted into FASTQ files using Illumina bcl2fastq conversion software v2.20.

Lexogen (Vienna, Austria) performed quality control, read processing and differential gene expression analysis up to the log fold change table: quality control analysis with fastqc, UMI extraction with umi_tools extract, trimming with cutadapt, STAR alignment to the reference genome of *S. pneumoniae* D39, umi deduplication with umi_tools dedup and differential gene expression analysis with DESeq2. With the differential gene expression table from Lexogen and by retrieving a STRING network setting the species to "Streptococcus pneumoniae D39" in Cytoscape 3.8.0, we created a gene association network with genes ≥1.25 log2FC upregulated or ≤−1.25 log2FC downregulated.

cDNA was synthesized using SuperScript IV Reverse transcriptase (Invitrogen), including RNaseOUT Recombinant Ribonuclease Inhibitor (Invitrogen), according to the manufacturer's protocol. RT-qPCR was done according to the manufacturer's protocol using TaqMan Fast Advanced Master Mix (Applied Biosystems) and Custom TaqMan Gene Expression Assays (Applied Biosystems) with sequences in Supplementary Table 2.

### Proteomic Analysis

Following overnight culture on agar plates, *S. pneumoniae* strain D39 was grown in BHI + 5% FCS to $OD_{600nm} = 0.5$ and 100 µl of overnight culture was sub-cultured in 10 ml of fresh CDM until $OD_{600nm} = 0.2$. The culture was split into two 5 ml cultures and 0.5 mg/ml peptide was added to one sample. After 15 min incubation at 37 °C, both cultures were centrifuged at $4000 \times g$ for 5 min at 4 °C. The pellet was washed in 1 ml PBS and transferred to an Eppendorf tube to centrifuge at $4000 \times g$ for 5 min. The supernatant was removed and the pellet was snap frozen in liquid nitrogen. Mass spectrometry was performed at the Core Facility Proteomics and Mass Spectrometry (University of Bern) as described as follows and elsewhere[58]. Bacterial pellets were lysed in 50 µl buffer containing 8 M urea and 100 mM Tris–HCl pH 8.0 and protease inhibitor cocktail (cOmplete w/o EDTA, Roche). Protein concentration was determined with Qubit assay (ThermoFisher) and adjusted to 1 mg/ml with lysis buffer and an aliquot corresponding to 10 µg was processed. Proteins were reduced by the addition of 1/10 volume of 0.1 M DTT and incubation for 30 min at 37 °C, followed by alkylation with a five-fold molar excess of iodoacetamide and incubation for 30 min at 37 °C in the dark. Iodoacetamide was quenched by 4/10 volume of 0.1 M DTT and urea concentration lowered to 4 M by addition of 20 mM Tris/HCl pH 8.0 with 2 mM $CaCl_2$. Proteins were digested for 2 h at 37 °C with sequencing grade LysC (Promega) at a protein/protease ratio of 50:1, followed by urea dilution to 1.6 M and overnight digestion at ambient temperature with 1:50 sequencing grade trypsin (Promega). Digestion was

stopped with trifluoroacetic acid (TFA) to a final concentration of 1% (v/v), followed by dilution to 0.1 μg/μl of protein.

Proteome analysis was done by nano-liquid chromatography on a Dionex Ultimate 3000 (Thermo Fisher Scientific) through a CaptiveSpray source coupling to a timsTOF Pro mass spectrometer (Bruker) with an end-plate offset of 500 V, a drying temperature of 200 °C, and with the capillary voltage fixed at 1.6 kV. A volume of 2 μl (200 ng) from the protein digest was loaded onto a pre-column (PepMap 100 C18, 5 μm, 100 A, 300 μm diameter × 5 mm length, ThermoFisher) at a flow rate of 10 μl/min with 0.05% (v/v) TFA in water/acetonitrile 98:2. After loading, peptides were eluted in back flush mode onto a home-made C18 CSH Waters column (1.7 μm, 130 Å, 75 μm × 20 cm) by applying a 90 min gradient of 5% acetonitrile to 40% in water/0.1% formic acid, at a flow rate of 250 nl/min and data was acquired in data-dependent acquisition mode using the parallel acquisition serial fragmentation (PASEF) option. The mass range was set between 100 and 1700 $m/z$, with 10 PASEF scans between 0.7 and 1.4 V s/cm². The accumulation time was set to 2 ms and the ramp time to 100 ms, respectively. Fragmentation was triggered at 20,000 arbitrary units, and peptides (up to a charge of 5) were fragmented using collision-induced dissociation with a spread between 20 and 59 eV. DDA data was interpreted and processed with MaxQuant (version 2.0.1.0) against the UniprotKB $S.$ $pneumoniae$ D39 strain protein sequences (release February 2021) as described elsewhere[58].

### Generation of human airway epithelial cell (hAEC) cultures
Human AEC cultures were cultured in 6.5 mm Transwells® with 0.4 μm pore polyester membrane insert in flat bottom 24-well plates (Corning, 3470) as described previously[21]. We seeded $1 × 10^5$ cells per insert. Experiments were performed once a pseudostratified layer of differentiated airway epithelial cells was formed, including cilia movement and physiological mucus production.

### Cytotoxicity assay in hAEC cultures
Cytotoxicity of 30 h peptide incubation on hAEC cultures was determined by measuring the release of lactate dehydrogenase (LDH), which is a stable cytosolic enzyme released upon cell lysis, and visualizing the ultrastructure of the epithelium. To the apical side of hAEC cultures, 40 μl of TEER solution (NaCl 0.9%, CaCl₂ 1.25 mmol/l and HEPES 10 mmol/l dissolved in distilled water) were added as negative control, 40 μl of peptide resuspended in TEER solution to 0.25 mg/ml and 0.5 mg/ml and 40 μl of 1x LDH+ control solution from the CytoTox® Non-Radioactive Cytotoxicity Assay (Promega) were added to the apical side. After 30 h of incubation, 160 μl TEER solution was added to the apical side and the medium from the apical and basolateral side were collected and both samples of each culture were used for cytotoxicity determination. Release of LDH was measured to determine the cytotoxicity of peptide V11A on hAEC cultures using the CytoTox® Non-Radioactive Cytotoxicity Assay (Promega) according to the manufacturer's instructions.

### Toxicity assay in zebrafish larvae
AB zebrafish eggs were kindly provided by Anna Gliwa, Institute of Anatomy (University of Bern), obtained by natural mating procedures. Zebrafish eggs were collected within the first hours post fertilization (hpf) and eggs and larvae were maintained at 28.5 °C on a 14 h/10 h light/dark cycle in a dedicated incubator (IPP110plus, Memmert GmbH) and used for experiments until no later than 5 days post fertilization (dpf). The fertilized eggs were kept for 1 day in E3 embryo medium (60× stock solution contains 34.8 g NaCl, 1.6 g KCl, 5.8 g CaCl₂, 9.78 g MgCl₂*6 H₂O) supplemented with 0.3 mg/ml methylene blue. On the following day, the media was changed to E3 only. Unfertilized eggs and deformed embryos were discarded. Peptides were resuspended in E3 to 0.5 mg/ml and 0.25 mg/ml. E3 was used as a negative control and 150 μg/ml LPS in E3 as a positive control for toxicity. Healthy larvae of similar developmental phases were picked 3 dpf and 15 larvae were transferred to each well of a six-well plate. For peptide toxicity testing, 2 wells (i.e. 30 larvae) were used per concentration, and for

positive and negative control, one well each. E3 was replaced by 2 ml of E3 without or with LPS or peptide and larvae were incubated for 48 h at 28.5 °C with light/dark cycle as above. At 48 h a DanioVision Observation Chamber (Nodus) dark/light protocol[59] with 1 h adaptation time, followed by 20 min dark, 10 min light, and 20 min dark, was run to investigate whether peptide treatment caused swimming behavioural changes with Ethovision XT 15 software. Larvae were euthanized after the experiment with 0.4% Tricaine.

### Adherence assay on hAEC cultures
Two days prior to infection, the basolateral medium of hAEC cultures was changed to a fresh medium without antibiotics. Bacterial culture of $S.$ $pneumoniae$ strain D39 was prepared as described for the growth assay. On the day of infection, 500 μl of the overnight culture were sub-cultured in 5 ml of fresh BHI until $OD_{600nm} = 0.4$, then centrifuged at 3000×$g$ for 7 min and the pellet was resuspended in 5 ml TEER solution for one wash, before resuspending again in 5 ml fresh TEER solution and diluting the bacterial solution for inoculation of hAEC cultures to $1 × 10^6$ cfu/ml. AEC cultures were washed with TEER to remove mucus at the time of infection. Peptides were resuspended in bacterial inoculum solution to 0.5 mg/ml and hAEC cultures inoculated with 40 μl of bacterial inoculum with or without peptides per insert (to give a final inoculum of $4 × 10^4$ cfu/insert). Next, the hAEC cultures were incubated for 24 h at 37 °C in a humidified atmosphere with 5% $CO_2$ then three washes with 200 μl of TEER solution were performed apically. Human AEC cultures were detached by adding 100 μl TrypLE™ Express Enzyme (1x), phenol red (Gibco) for 30 min at 37 °C, 5% $CO_2$ with resuspension after the first 10 min before continuing the incubation time. The trypsinized samples were collected into 100 μl of BHI with 5% FCS and directly processed by making dilutions and plating them onto CSBA plates to determine cfu after overnight incubation at 37 °C, 5% $CO_2$.

### Rat colonization model
Bacterial culture of $S.$ $pneumoniae$ strain 1154.75 (serotype 23F) was prepared as described for the growth assay. On the day of inoculation, 500 μl of the overnight culture were sub-cultured in 5 ml of BHI until $OD_{600nm} = 0.4$, then centrifuged at 3000×$g$ for 7 min and the pellet washed in 0.85% NaCl solution before resuspending again, in 5 ml 0.85% NaCl solution for an inoculum of $5 × 10^7$ cfu/ml. We performed two experiments on different occasions: (1) two litters each of 12 (6 male, 6 female) nursing Wistar rats along with their dams (specific pathogen-free, SPF) were obtained from Charles River (Germany), kept in individually ventilated cages with a 12 h light/dark cycle and constant temperature of 22 ± 2 °C, and provided with tap water and gamma-sterilized pellet diet ad libitum. Once 14 days old, the rat pups were inoculated intranasally with the bacterial inoculum with or without 0.5 mg/ml peptide V11A by dripping 5 μl of inoculum into each nare, in alternate. Separate litters of 12 pups were used for no peptide and peptide conditions. Then, 24 h after bacterial inoculation, 6 pups of each litter (3 male, 3 female) were sacrificed by an overdose of pentobarbital (150 mg/kg, i.p.). Nasal wash was done by flushing through the trachea with 200 μl 0.85% NaCl solution and harvesting from the nostrils, and the nasopharynx was extracted. For the remaining pups, 0.85% NaCl solution or 0.5 mg/ml V11A in 0.85% NaCl solution was administered again and 48 h after bacterial inoculation, sacrifice, nasal wash and nasopharynx extraction were done as above. (2) The second experiment was done similarly to the first but with the following changes: one litter of 10 (5 male, 5 female) nursing Wistar rats along with their dam were obtained and kept in a ventilated cage. Once 14 days old, the rat pups were inoculated intranasally with the bacterial inoculum with or without 0.5 mg/ml peptide V11A by dripping 5 μl of inoculum into each nare, in alternate. All pups were kept in the same cage. Then, 24 h after bacterial inoculation, all pups were sacrificed and nasal wash and nasopharynx extraction were performed as above. The extracted nasopharynxes were placed into 2 ml homogenization tubes with beads (2.8 mm diameter ceramic beads, BER0072, Labgene Scientific) containing 300 μl of PBS on ice and homogenized with 3 cycles of 30 s and 4500 rpm and 30 s break with Precellys Evolution homogenizer (Bertin

Technologies). The nasal wash and nasopharynx samples were directly processed by making dilutions and plating them onto CSBA plates with 3 µg/ml trimethoprim and 57 µg/ml sulfamethoxazole to determine cfu after overnight incubation of the plates at 37 °C, 5% $CO_2$.

## Statistics

All analyses were done in the R version 4.0.3 within R studio version 1.3.1093. Shapiro–Wilk normality tests were used to evaluate normality. If more than one sample group was not normally distributed, the Wilcoxon rank sum test was done, otherwise $t$-test was used. As no cell of the expected frequencies in the contingency table was less than 5, a Chi-squared test was performed to test for the effect of peptide V11A on number of rats colonized. In all analyses, a $p$-value of $\leq 0.05$ was considered statistically significant.

## Ethics

All research was performed in accordance with the relevant guidelines and regulations.

In accordance with the Swiss Human Research Law, researchers at Bern University Hospital received anonymized discarded leftovers of human cerebrospinal fluid samples collected in clinical routine. For the human primary airway epithelial cell model the anonymized post-mortem tissue material was obtained through the Tissue Bank Bern (TBB) in accordance with ethical approval (KEK-BE 1571/2019).

Studies with the rats were approved by the Animal Care and Experimentation Committee of the Canton Bern, Switzerland, and follow the ethical principles and guidelines for experiments on animals as published by the Swiss Federal Veterinary Office, licence number BE64/2022.

## Reporting summary

Further information on research design is available in the Nature Portfolio Reporting Summary linked to this article.

## Data availability

All data generated or analysed during this study are included in this published article and supplementary files. The source data underlying the figures and Supplementary figures can be found in Supplementary Data 1. The transcriptomic data for this study has been deposited in the European Nucleotide Archive (ENA) at EMBL-EBI under accession number PRJEB73628 and processed data (normalized counts and differential gene expression analysis) are available in Supplementary Data 2. The mass spectrometry proteomics data have been deposited to the ProteomeXchange Consortium via the PRIDE[60] partner repository with the dataset identifier PXD050408 and processed data are available in Supplementary Data 3.

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

## Acknowledgements

We are grateful to the following people for providing pneumococcal strains: Jeffrey Weiser (New York University, USA) for D39, Philippe Moreillon (Centre Hospitalier Universitaire Vaudois, Switzerland) for R6 and Ralf Reinert (National Reference Center for Pneumococcus, Germany) for reference strains. We thank Carlo Casanova (Institute for Infectious Diseases, University of Bern) for kindly providing strains of *S. pneumoniae, H. influenzae, S. aureus, M. catarrhalis* and *K. pneumoniae*. We would like to thank the Next Generation Sequencing Platform, University of Bern, for performing the high-throughput sequencing experiments. We are grateful to Sabina Berezowska and Irene Ramos-Centeno (Institute of Pathology, University of Bern) for providing the tissues for the hAEC culture establishment via the Tissue Bank Bern. We thank Anna Gliwa (Institute for Anatomy, University of Bern) for providing zebrafish larvae eggs. We are grateful to Marianne Küffer, Adrian Wegmüller and Linus Rechsteiner (Institute for Infectious Diseases, University of Bern), Sophie Braga-Lagache and

Natasha Buchs (Proteomics and Mass Spectrometry Core Facility, University of Bern) and Zhian Salehian (Norwegian University of Life Sciences) for their technical support. This work was funded by grant number 192067 from the Swiss National Science Foundation to L.J.H. We gratefully acknowledge the Swiss National Science Foundation (R'Equip grant no. 316030-189737) and the University of Bern for financing the mass spectrometry equipment.

## Author contributions

L.J.H. conceived the study, J.L. wrote the first draft of the manuscript. J.L., L.J.H., D.G. and S.L.L. participated in the design of the study. J.L., H.P. and L.S.G. did bacterial growth assays. D.S. provided D39 parental strain and D39Δ*amiC* and R.H. provided hCSF. J.L. did epifluorescence microscopy after preliminary experiments and establishment of a protocol with C.G. and J.W.V. J.L. analysed bacterial cell shape, chain length, transformation rate, RNA-Seq and proteomics. N.J.C. provided support for the analysis of RNA-Seq data in R. M. He. analysed mass spectrometry data. J.L., L.L., M.F.L. and R.D. performed or advised on hAEC culture assays. L.H. did cytotoxicity assay with zebrafish larvae. J.L., M.E. and D.G. performed experiments with infant rats. All authors gave final approval for publication.

## Competing interests

The authors declare the following competing interests: L.J.H. and J.L. are inventors on a patent application pertaining to this work. R.H. received speaker/advisor honorary from Merck, Novartis, Roche, Biogen, Alexion, Sanofi, Janssen, Bristol-Myers Squibb, Teva/Mepha and Almirall. He received research support within the last 5 years from Roche, Merck, Sanofi, Biogen, Chiesi, and Bristol-Myers Squibb. He also received research grants from the Swiss MS Society and the SITEM Insel Support Fund and is a member of the Advisory Board of the Swiss and International MS Society. He also serves as deputy editor-in-chief for the Journal of Central Nervous System Disease. All conflicts are not related to this work. All other authors declare no competing interests.
