## [Peer review file · Communications Biology]

Reviewers' comments:

Reviewer #1 (Remarks to the Author):

Lux et al present a fascinating study describing the antibacterial effect of V11A a small peptide isolated from *Klebsiella pneumoniae*. They assess the effect of this peptide on the pathogen *Streptococcus pneumoniae*, a major public health threat globally for which antimicrobial resistance is rising and thus alternatives for treatment are needed. They demonstrate that V11A impedes pneumococcal growth and that this is mediated by uptake through AmiA/AlmA/AlmB oligopeptide permease, leading to an altered transcriptome and proteome, as well as other effects including impeded adherence and colonisation in animal model.

Overall, I found that this was a well conducted study. Most experiments included appropriate controls and authors use a number of approaches to verify their findings and show the effects they observe are due to the action of the V11A peptide. I found it enjoyable to read this manuscript.

I do have some comments for the authors to address/consider:

1. Throughout the manuscript there are a few claims of V11A being used as 'therapeutic'. At this stage, it is inappropriate to make such a claim, but rather it demonstrates strong potential to be used as therapeutic. For example, the abstract states "we propose a potential novel treatment for pneumococcal disease" (line 37). However, V11A was only tested in an asymptomatic carriage model, not disease and was delivered simultaneously with pneumococcus, not therapeutically. Similarly, line 297 claims V11A has "superior therapeutic potential" for suppressing growth, but it was not clearly explained anywhere whether V11A was added at the time the growth curve commenced (which I assume is the case, but would not be therapeutic) or after growth had commenced.
2. Some clarification on the fluorescence microscopy results (Figure 2) are needed. The authors claim "We found clear homogeneous FITC staining intracellularly only in the D39 strain with a functional permease and not in the Δ amiC mutant" (line 117-118). However, I can see a weak fluorescent signal in the mutant pneumococci. Is this simply background signal? Or could it be that there is still uptake in the mutant, just reduced uptake (as opposed to no uptake as claimed)? Some information clarifying this point (or changing the text to say it is reduced uptake in the mutant, not no uptake) is needed.
3. The pneumococcal colonisation data from the rat model was analysed by comparing bacterial load data. Although this is informative (and should be kept in the manuscript), this is a more indirect approach of analysing colonisation and perhaps not appropriate given most of the bacterial load datapoints in the V11A group were 0 or below the limit of detection. Colonisation is binary – rats are either colonised by pneumococci or they are not. Therefore, if the authors wish to claim V11A prevent colonisation (as they state throughout the manuscript), it would be more appropriate to analyse by 'proportion of rats colonised'. As hypothetical example 10/10 (100%) vs 2/10 (20%) of rats having detectable pneumococci in the nasopharynx, in control vs treated conditions, respectively which can be analysed by a Fishers exact test.
4. The discussion could benefit a little more text explaining the isolates with the mutations in AmiA that meant they were not affected by V11A. How prevalent is this mutation in the pneumococcal population? There is a brief mention in line 309 but more further detail would be beneficial (e.g. how many genomes did your BLAST look at?). This would be an important consideration for the use of V11A as a therapeutic agent if it is a common mutation of V11A can induce selective pressure to mutate. Similarly, this seems to contradict claims later in the discussion (Line 308) that say mutations in AmiA and AlmA are less likely to occur due to loss of function – but the authors had two isolates with mutations in AmiA already so clearly this is possible??
5. When examining the specific amino acids in the peptide that are important for the antibacterial effect, the authors make claims that it has "substitutions" or a "switch" from other species (line 87). This is an inappropriate conclusion to draw as it implies the direction of the changes i.e. the ancestral *Klebsiella* peptide had these residues and then mutated into the current V11A form. There is not enough evidence to infer a direction of mutation or evolution in either direction. Instead, the authors should state that these are simply different amino acids at these key positions. They can claim substitution for the peptides they created because they introduced those changes themselves.
6. Remove personal interpretations from figure legends. Figures and figure legends are a

representation of the data and the details required for the reader to understand the figure/data so their own interpretations can be made. They should remain unbiased of the authors interpretations/opinions. Interpretations and opinions of the authors should be in the manuscript (results and discussion), not in the figure legends.

7. Remove lines 316-317 – this is too speculative. As no information on the Klebsiella strain used in this study is provided and no information on how prevalent V11A is in Klebsiella strains, it is not appropriate to claim this is a mechanism Klebsiella uses to gain advantage of pneumococcus.

Likewise, there is no proof the Klebsiella V11A has evolved to target pneumococcus. It is equally plausible that V11A is effective against pneumococcus just by chance.

8. I assume swimming behaviour is a well-established metric for measuring toxicity in zebrafish? Please provide references to support this. Otherwise, it is unclear how the authors determined that this phenotype can indicate toxicity.

Reviewer #2 (Remarks to the Author):

In this study, Lux and al identified a peptide secreted by Klebsiella pneumoniae that inhibit the growth and colonization of the human pathogen Streptococcus pneumoniae. The authors found that the peptide opportunistically enters S. pneumoniae via an oligopeptide ABC importer and subsequently induces pleiotropic biological changes that the authors extensively characterized by a large range of methodologies. Uptake of the peptide V11A led to the alteration of gene expression, caused irregular cell shape, decreased chain length and decreased genetic transformation. The bacteriostatic effect of the peptide was specific to S. pneumoniae and S. pseudopneumoniae and depended on peptide sequence and length. Furthermore, the peptide reduced pneumococcal adherence to human human airway epithelial cell cultures and colonization of rat nasopharynx, without apparent toxicity in a zebrafish larvae model.

Although the effects of V11A are visible at relatively high doses, and despite the fact that the precise cellular target(s) of V11A is (are) still unknown, this work is original and provides compelling evidence that K. pneumoniae synthesizes a peptide that can inhibit the growth cycle of S. pneumoniae by hijacking one of its ABC transporter to enter the cell. The paper is well written and I genuinely enjoyed reading the manuscript. I recommend to accept the paper after minor revisions

1) The authors analyzed by LC-MS/MS the secretome of K. pneumoniae but did not provide a list of identified peptides. Rather, they only revealed the presence of 3 peptides containing the sequence VNATDEDRWNA. It would be informative for the community if the authors were willing to share their results

2) V11A affected pneumococcal morphology, chain length and transformation (Fig. 3). Notably, the peptide quite spectacularly reduced the transformation rate of S. pneumoniae (by 21-fold). However the experiment in Fig. 3c is lacking a control. Could the authors add an experimental condition with CSP-1+V11A mutant (at position 4 or 10) ?

3) It would be also useful to include a V11A mutant for the colonization experiment in Fig. 5b. One can observe a large response variability in the « none » condition.

4) Optional. It would be nice to biochemically show a direct interaction between the peptide and purified AmiA. Substrate-binding proteins of ABC transporters are usually fairly easy to purify.

Response to Reviewers

Ref: Submission COMMSBIO-23-4507-T

"Klebsiella pneumoniae peptide hijacks a Streptococcus pneumoniae permease to subvert pneumococcal growth and colonization"

Janine Lux, Hannah Portmann, Lucía Sánchez García, Maria Erhardt, Lalaina Holivololona, Laura Laloli, Manon F. Licheri, Clement Gallay, Robert Hoepner, Nicholas J. Croucher, Daniel Straume, Jan-Willem Veening, Ronald Dijkman, Manfred Heller, Denis Grandgirard, Stephen L. Leib, Lucy J. Hathaway

We thank both reviewers for their helpful comments which have enabled us to improve the manuscript as described in the point-by-point responses in the table below and as tracked changes in the revised manuscript.

Reviewer #1 (Remarks to the Author):

Lux et al present a fascinating study describing the antibacterial effect of V11A a small peptide isolated from Klebsiella pneumoniae. They assess the effect of this peptide on the pathogen Streptococcus pneumoniae, a major public health threat globally for which antimicrobial resistance is rising and thus alternatives for treatment are needed. They demonstrate that V11A impedes pneumococcal growth and that this is mediated by uptake through Ami-AliA/AliB oligopeptide permease, leading to an altered transcriptome and proteome, as well as other effects including impeded adherence and colonisation in animal model.

Overall, I found that this was a well conducted study. Most experiments included appropriate controls and authors use a number of approaches to verify their findings and show the effects they observe are due to the action of the V11A peptide. I found it enjoyable to read this manuscript.

I do have some comments for the authors to address/consider:

Reviewer comments	Response
1. Throughout the manuscript there are a few claims of V11A being used as ‘therapeutic’. At this stage, it is inappropriate to make such a claim, but rather it demonstrates strong potential to be used as therapeutic. For example, the abstract states “we propose a potential novel treatment for pneumococcal disease” (line 37). However, V11A was only tested in an asymptomatic carriage model, not disease and was delivered simultaneously with pneumococcus, not therapeutically.	We agree and have changed the text in the abstract, line 37, to "We identified a peptide with potential as a therapeutic for pneumococcal diseases" and removed the claim previously on line 297.

Similarly, line 297 claims V11A has “superior therapeutic potential” for suppressing growth, but it was not clearly explained anywhere whether V11A was added at the time the growth curve commenced (which I assume is the case, but would not be therapeutic) or after growth had commenced.	
2. Some clarification on the fluorescence microscopy results (Figure 2) are needed. The authors claim “We found clear homogeneous FITC staining intracellularly only in the D39 strain with a functional permease and not in the ΔAmiC mutant” (line 117-118). However, I can see a weak fluorescent signal in the mutant pneumococci. Is this simply background signal? Or could it be that there is still uptake in the mutant, just reduced uptake (as opposed to no uptake as claimed)? Some information clarifying this point (or changing the text to say it is reduced uptake in the mutant, not no uptake) is needed.	We accept this point and have changed the text to "greatly reduced uptake" (now line 131).
3. The pneumococcal colonisation data from the rat model was analysed by comparing bacterial load data. Although this is informative (and should be kept in the manuscript), this is a more indirect approach of analysing colonisation and perhaps not appropriate given most of the bacterial load datapoints in the V11A group were 0 or below the limit of detection. Colonisation is binary – rats are either colonised by pneumococci or they are not. Therefore, if the authors wish to claim V11A prevent colonisation (as they state throughout the manuscript), it would be more appropriate to analyse by ‘proportion of rats colonised’. As hypothetical example 10/10 (100%) vs 2/10 (20%) of rats having detectable pneumococci in the nasopharynx, in control vs treated conditions, respectively which can be analysed by a Fishers exact test.	We thank Reviewer 1 for this suggestion and have incorporated it into the new version of the manuscript. In addition, we have performed another in vivo experiment for the 24h timepoint to have more data and have analyzed the number and % of rats colonized as well as the bacterial load. The extra data enabled us to perform a Chi-squared test for binary analysis of colonization. The text, line 226, now reads "At 24 h pneumococci were detected in the nasopharynx of 9 out of 11 rats (81.81 %) in the absence of V11A and in 3 out of 11 (27.27 %) in the nasopharynx of rats which had received V11A (Chi-squared test: p = 0.03228)." As we combined the data of the new experiment with that of the original Fig. 5b, we have updated Fig. 5b, Supplementary Fig. 7 and the corresponding text for methods, number of animals and statistical results.
4. The discussion could benefit a little more text explaining the isolates with the mutations in AmiA that meant they were not affected by V11A. How prevalent is this mutation in the pneumococcal population? There is a brief	The following text has been added to the Results, beginning on line 116: "A BLAST search of the AmiA sequence covering the relevant mutation was performed to determine its frequency: YGYVYTADPETLDYLISRK (mutation in bold and

mention in line 309 but more further detail would be beneficial (e.g. how many genomes did your BLAST look at?). This would be an important consideration for the use of V11A as a therapeutic agent if it is a common mutation of V11A can induce selective pressure to mutate. Similarly, this seems to contradict claims later in the discussion (Line 308) that say mutations in AmiA and AliA are less likely to occur due to loss of function – but the authors had two isolates with mutations in AmiA already so clearly this is possible??	underlined). Of 5000 sequences examined, only 4 (0.08%) had this mutation." and to the Discussion beginning on line 344: "and the mutation in AmiA of strain R6, associated with lack of response to V11A, was found in fewer than 0.08% of pneumococcal genomes."
5. When examining the specific amino acids in the peptide that are important for the antibacterial effect, the authors make claims that it has “substitutions” or a “switch” from other species (line 87). This is an inappropriate conclusion to draw as it implies the direction of the changes i.e. the ancestral Klebsiella peptide had these residues and then mutated into the current V11A form. There is not enough evidence to infer a direction of mutation or evolution in either direction. Instead, the authors should state that these are simply different amino acids at these key positions. They can claim substitution for the peptides they created because they introduced those changes themselves.	We agree with this comment and have changed "substitution" and "switch" to "differences" through the text, as highlighted in tracked changes and also in the figure legends.
6. Remove personal interpretations from figure legends. Figures and figure legends are a representation of the data and the details required for the reader to understand the figure/data so their own interpretations can be made. They should remain unbiased of the authors interpretations/opinions. Interpretations and opinions of the authors should be in the manuscript (results and discussion), not in the figure legends.	We have made this change in the legends of all figures, table 1, supplementary figures 2 -7 and supplementary table 1.
7. Remove lines 316-317 – this is too speculative. As no information on the Klebsiella strain used in this study is provided and no information on how prevalent V11A is in Klebsiella strains, it is not appropriate to claim this is a mechanism Klebsiella uses to	We agree and have removed this. The text now reads, starting on line 352: "We speculate that import of the peptide via the pneumococcal AmiA-AliA/AliB permease may be advantageous to K. pneumoniae in the shared niche of the nasopharynx."

gain advantage of pneumococcus. Likewise, there is no proof the Klebsiella V11A has evolved to target pneumococcus. It is equally plausible that V11A is effective against pneumococcus just by chance.	
8. I assume swimming behaviour is a well-established metric for measuring toxicity in zebrafish? Please provide references to support this. Otherwise, it is unclear how the authors determined that this phenotype can indicate toxicity.	We have added references 23 and 24 and cited them in line 206: "This light-dark locomotion test is a well-established metric for toxicity measurements which can reveal, for example, reactivity and muscular weakness ^{23, 24} ."

Reviewer #2 (Remarks to the Author):

In this study, Lux and al identified a peptide secreted by Klebsiella pneumoniae that inhibit the growth and colonization of the human pathogen Streptococcus pneumoniae. The authors found that the peptide opportunistically enters S. pneumoniae via an oligopeptide ABC importer and subsequently induces pleiotropic biological changes that the authors extensively characterized by a large range of methodologies. Uptake of the peptide V11A led to the alteration of gene expression, caused irregular cell shape, decreased chain length and decreased genetic transformation. The bacteriostatic effect of the peptide was specific to S. pneumoniae and S. pseudopneumoniae and depended on peptide sequence and length. Furthermore, the peptide reduced pneumococcal adherence to human human airway epithelial cell cultures and colonization of rat nasopharynx, without apparent toxicity in a zebrafish larvae model. Although the effects of V11A are visible at relatively high doses, and despite the fact that the precise cellular target(s) of V11A is (are) still unknown, this work is original and provides compelling evidence that K. pneumoniae synthesizes a peptide that can inhibit the growth cycle of S. pneumoniae by hijacking one of its ABC transporter to enter the cell. The paper is well written and I genuinely enjoyed reading the manuscript. I recommend to accept the paper after minor revisions

Reviewer comments	Response
1) The authors analyzed by LC-MS/MS the secretome of K. pneumoniae but did not provide a list of identified peptides. Rather, they only revealed the presence of 3 peptides containing the sequence VNATDEDRWNA. It would be informative for the community if the authors were willing to share their results	We are characterizing other peptides from the secretome and intend to share the results with the community when this work is finished by publishing this in the future.
2) V11A affected pneumococcal morphology, chain length and transformation (Fig. 3). Notably, the peptide quite spectacularly	We thank Reviewer 2 for this suggestion and have now repeated the transformation experiment an additional 5 times, including a control with CSP-

reduced the transformation rate of S. pneumoniae (by 21-fold). However the experiment in Fig. 3c is lacking a control. Could the authors add an experimental condition with CSP-1+V11A mutant (at position 4 or 10) ?	1+V11A_mut4. This data has been combined with that of the original Fig. 3c to produced the new Fig. 3c. We have updated the figure legend and text accordingly, line 149: "V11A, but not the control peptide (V11A with amino acid difference at position 4) reduced the mean transformation rate from 1.44 % to 0.169 %, which equals a 8.5-fold reduction (Fig. 3c)".
3) It would be also useful to include a V11A mutant for the colonization experiment in Fig. 5b. One can observe a large response variability in the « none » condition.	We agree that it would be interesting to include a V11A mutant control group in the colonization experiment. However, having two control groups raised ethical concerns about minimizing the number of animals used. However, we were able to perform an additional colonization experiment for the 24 h timepoint with and without V11A peptide. This data has been combined with that of the original Fig. 5b to give the new Fig. 5b and the data in Supplementary Fig. 7. The figure legends and text have been updated accordingly.
4) Optional. It would be nice to biochemically show a direct interaction between the peptide and purified AmiA. Substrate-binding proteins of ABC transporters are usually fairly easy to purify.	We agree that it would be interesting to show this and have tried to do it by tryptophan binding assay but unfortunately we found that the peptide has too much intrinsic fluorescence to be used in this assay.

REVIEWERS' COMMENTS:

Reviewer #1 (Remarks to the Author):

The authors have made considerable effort to address my concerns, for which I thank them. I am satisfied with their revised manuscript and do not have any further concerns.

Reviewer #2 (Remarks to the Author):

I am satisfied with the changes of the revised manuscript. As a note, i feel that the authors did not appropriately understand and respond to the fifth comment of the other reviewer. The authors should keep "substitutions" when mutations were introduced by peptide synthesis. For instance, line 93 should be written with "differences", while lines 94 and 95 should be written with "substitutions". But this can be easily fixed at the proof stage, and i congratulate the authors for their work.